# High endemicity of alveolar echinococcosis in Yili Prefecture, Xinjiang Autonomous Region, the People's Republic of China: Infection status in different ethnic communities and in small mammals

**Baoping Guo[1]◉, Zhuangzhi Zhang[2]◉, Yongzhong Guo[3]◉, Gang Guo[1], Haiyan Wang[4], Jianjun Ma[5], Ronggui Chen[6], Xueting Zheng[1], Jianling Bao[1], Li He[1], Tian Wang[1], Wenjing Qi[1], Mengxiao Tian[1], Junwei Wang[2], Canlin Zhou[1], Patrick Giraudoux[7], Christopher G. Marston[8], Donald P. McManus[9], Wenbao Zhang[1]*, Jun Li[1]***

**1** State Key Laboratory of Pathogenesis, Prevention and Treatment of High Incidence Diseases in Central Asia, Xinjiang Medical University, and WHO-Collaborating Centre for Prevention and Care Management of Echinococcosis, The First Affiliated Hospital of Xinjiang Medical University, Urumqi, Xinjiang, China, **2** Veterinary Research Institute, Xinjiang Academy of Animal Sciences, Urumqi, Xinjiang, China, **3** The Friendship Hospital of Yili Kazak Autonomous Prefecture, Yining, Xinjiang, China, **4** Chabuchaer Center for Disease Control and Prevention, Chabuchaer, Xinjiang, China, **5** Xinyuan Center for Disease Control and Prevention, Xinyuan, Xinjiang, China, **6** Yili Center for Animal Disease Control and Prevention, Yining, Xinjiang, China, **7** Chrono-environment lab, UMR6249, University of Franche-Comte and CNRS, Besancon, France, **8** Land Use Group, UK Centre for Ecology and Hydrology, Lancaster, United Kingdom, **9** Molecular Parasitology Laboratory, Infectious Diseases Program, QIMR Berghofer Medical Research Institute, Brisbane, Queensland, Australia

◉ These authors contributed equally to this work.
* wenbaozhang2013@163.com (WZ); 1742712944@qq.com (JL)

## Abstract

Alveolar echinococcosis (AE) is a life-threatening disease in humans caused by the larval stage of *Echinococcus multilocularis*. The tapeworm is transmitted between small mammals and dogs/foxes in the Northern Hemisphere. In this study 286 AE cases were reported from eight counties and one city in Yili Prefecture, Xinjiang Autonomous Region, the People's Republic of China from 1989 to 2015 with an annual incidence (AI) of 0.41/100,000. Among the patients, 73.08% were diagnosed in the last 11 years. Four counties in the high mountainous areas showed higher AI (0.51–1.22 cases/100,000 residents) than the four counties in low level areas (0.19–0.29/100,000 residents). The AI of AE in Mongolian (2.06/100,000 residents) and Kazak (0.93/100,000 residents) ethnic groups was higher than the incidence in other ethnic groups indicating sheep-farming is a risk for infection given this activity is mainly practiced by these two groups in the prefecture. A total of 1411 small mammals were captured with 9.14% infected with *E. multilocularis* metacestodes. *Microtus obscurus* was the dominant species in the mountain pasture areas with 15.01% of the voles infected, whereas *Mus musculus* and *Apodemus sylvaticus* were the dominant small mammals in the low altitude areas. Only 0.40% of *A. sylvaticus* were infected with *E. multilocularis*. PCR amplification and sequencing analysis of the mitochondrial *cox1* gene showed that *E. multilocularis* DNA sequences from the small mammals were identical to isolates of local human

**Data Availability Statement:** All relevant data are within the manuscript and its supporting information files.

**Funding:** This study was supported by Natural Science Foundation of China (CN) (81830066 for WZ and U1803282 for JL). The funders had no role in study design, data collection and analysis, decision to publish, or preparation of the manuscript.

**Competing interests:** The authors have declared that no competing interests exist.

AE cases. The overall results show that Yili Prefecture is a highly endemic area for AE and that the high-altitude pasture areas favorable for *M. obscurus* may play an important role in its transmission in this region.

## Author summary

Alveolar echinococcosis (AE) is a neglected zoonosis caused by the larval stage of the fox/dog tapeworm *Echinococcus multilocularis*. In this study, we collected data on 286 AE cases reported from Yili Prefecture, Xinjiang Autonomous Region, the People's Republic of China from 1989 to 2015 with an annual incidence (AI) of 0.41/100,000. Among the patients, 73.08% were diagnosed in the last 11 years. The incidence (0.51–1.22 cases/100,000 residents) was higher in the high-altitude mountainous areas than those in low level areas (0.19–0.29/100,000 residents). In term of ethnic group, the AI of AE in Mongolian (2.06/100,000 residents) and Kazak (0.93/100,000) groups had higher incidence than the other ethnic groups, indicating sheep-farming activity is a risk for infection given that sheep farming is mainly practiced by these two groups in the prefecture. A total of 1411 small mammals were captured with 9.14% infected with *E. multilocularis* metacestodes. *Microtus obscurus* was the dominant species captured in the mountainous pasture areas with 15.01% infection rate, whereas *Mus musculus* and *Apodemus sylvaticus* were the dominant small mammals in the low altitude areas. Only 0.40% of *A. sylvaticus* were infected with *E. multilocularis*. These findings show that Yili Prefecture is a highly endemic area for AE and that the high-altitude pasture areas favorable for *M. obscurus* may play an important role in its transmission in this region.

## Introduction

*Echinococcus multilocularis* infection causes alveolar echinococcosis (AE) in humans. This lethal disease is endemic in the Northern Hemisphere [1,2] with a median of 666,434 DALYs (disability-adjusted life years) per annum resulting [3]. The life-cycle of *E. multilocularis* involves two kinds of mammalian host species, intermediate hosts including small mammals such as *Microtus* rodents, and canid definitive hosts including foxes (*Vulpes vulpes*), dogs (*Canis familiaris*), and the raccoon dog (*Nyctereutes procyonoides*) [4]. Definitive hosts are infected by predating small mammals that harbor *E. multilocularis* metacestodes containing protoscoleces (PSCs). Approximately 30 days post-infection, the PSCs develop into mature tapeworms producing eggs in the intestine of the definitive host [5]. The eggs are released into the environment in the feces of the definitive hosts, and subsequently infect small mammals via oral ingestion when feeding on vegetation. Thus, *E. multilocularis* transmission typically occurs in a cycle between foxes/dogs and small mammals [2]; humans can also be infected, although they are accidental hosts not directly involved in the life-cycle of the parasite [1].

In the People's Republic of China, AE is highly endemic, particularly on the Qinghai-Tibet Plateau [3,4,6–9]; Kazakhstan and Kyrgyzstan are also highly endemic for the disease [3]. Xinjiang Autonomous Region bridges these two endemic areas and is, historically, an AE-endemic area [10]. However, the recent endemic situation in this region is unclear, with no updated reports since 2000 [10]. In this study, we retrospectively collected human AE cases from eight counties and one city in Yili Prefecture, which shows that AE incidence increased over the 20 year period. Our surveys of *E. multilocularis* infection in small mammals indicated that Yili

Prefecture is a natural focus of *E. multilocularis*, and the resulting AE cases in human communities from this region may be associated with the distribution of *Microtus spp.* small mammals and animal farming activity.

## Materials and methods

### Ethics statement

This study, including the capture of wild small mammals during field survey, was approved by the Ethics Committee of the First Affiliated Hospital of Xinjiang Medical University, Urumqi, China (approval number 20140619–12). All patients involved in the study also provided their written informed consent including child patients with the written informed consents from their parents/guardians.

### Study area

Yili Prefecture is located in the west Tianshan Mountains in Xinjiang Autonomous Region, The People's Republic of China, bordering Kazakhstan. The prefecture has eight counties and one city, Yining City, and is traversed by the Yili River which is itself formed by the convergence of three main upstream rivers, the Kashi River, Kunus River and Tekes River (Fig 1). There are five counties in the high altitude mountainous pasture area (HAPA) in the upstream regions of the Yili River, these being Nileke, Xinyuan, Gongliu, Tekes and Zhaosu Counties. Yining City and Yining, Chabuchaer and Huocheng counties are located downstream in the low altitude mountainous agricultural area (LAAA) of the Yili River. In 2015, Yili Prefecture (8 counties and 1 city) had a total population of 2.69 million with the major ethnic groups being Kazak, Han, Uygur, Hui/Dongxiang, Mongol, Xibo and Uzbek. In the HAPA there are approximately 20 million hectares pasture areas higher than 1800 meters above sea level that are utilized for farming, supporting a total of 6.688 million domestic animals including sheep, cattle and horses [11]. The annual average temperature in the HAPA is 2.9–5.9˚C with an average rainfall of 600 mm. The down-stream LAAA region comprises 0.49 million hectares of farmland with an annual average temperature of 9.1–11.1˚C and annual average rainfall of 417.6 mm.

### Data collection and analysis in Yili Prefecture

All AE cases reported in this study were registered in four hospitals with approved surgical teams and facilities for treating this disease. These are the Yili Friendship Hospital, Xinhua Hospital, Nongsishi Hospital and the First Affiliated Hospital of Xinjiang Medical University. These are the only hospitals/clinics for treating AE in this region. The duration period of the study was from 1989 (the time of the first confirmed case) to 2015. All AE patients were diagnosed by radiology, serology, and/or pathological testing of tissue sections following surgical removal of lesions, and all cases were treated with albendazole. A state-driven control program was commenced in 2005 [2]. An average annual incidence (AI) was calculated based on cumulative incidence [12] for comparison of AE infectious status over two periods (1994–2004 and 2005–2015) in Yili Prefecture. We collected population statistics from the Sixth National Census [http://blog.sina.com.cn/s/blog_4b0a5e8a0101cc2o.html] and Yili Annual Reports (YILI-HASAKEZIZHIZHOU TONGJINIANJIAN) from 1999 to 2015, as released by the Yili Bureau of Statistics.

### Small mammal collection and *E. multilocularis* detection

Three counties were selected for small mammal sampling in the HAPA (Nileke, Xinyuan and Tekes Counties) with one further county, Chabuchaer County, selected in the LAAA. In each

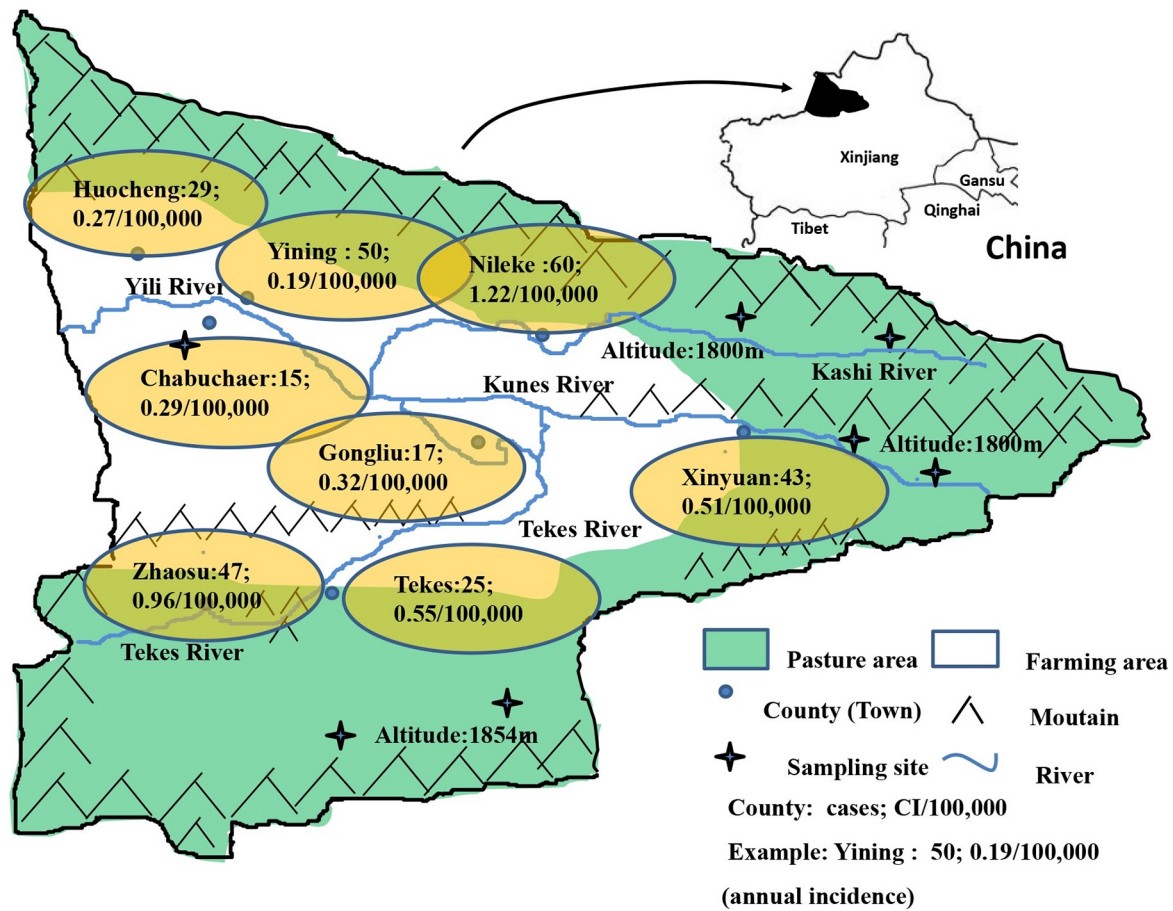

**Fig 1. Distribution of alveolar echinococcosis cases in Yili Prefecture, Xinjiang.** In the figure, "green" indicates grassland, "white" indicates agricultural land, "yellow" indicates the city name and location, "cross" indicates the sampling site, and "blue curve" indicates a river. Cumulative incidences in each of the counties are also indicated.

selected HAPA county, two different locations were selected (Fig 1). To capture small mammals in the HAPA locations, water was pumped into burrows to drive out the small mammals, which were subsequently captured. In Chabuchaer County, small mammals were captured using trapping methods [13]. Snap traps 10×15 cm in size were placed linearly along the edge of agricultural farmland with 10 meter intervals between traps. Over a period of 3 weeks in August/September, 2017, 200 traps were laid each afternoon and these were collected on the afternoon of the following day.

The livers of the captured small mammals were visually inspected for *E. multilocularis* metacestodes by three staff members. Suspected infected livers displaying white dots, or cyst like lesions or tissues with unusual surface appearances, were fixed with 75% (v/v) ethanol and stored at 4˚C until parasite and small mammal identification was undertaken. Non-*Microtus* species were identified by morphological characteristics.

## Haematoxylin and eosin (H&E) staining

All small mammal livers suspected of AE infection were sectioned for histological examination. Twenty non-infected livers were sectioned and stained as negative controls. Each liver was cut into three sections and dehydrated using a Leica automatic dehydrator. The specimens

**Table 1. Primers designed for amplification of the partial sequences of *E. multilocularis* *cox1*, *nad2* and *cob* genes using nested PCR.**

| Primer | Sequence (5′-3′) (First amplification) | Sequence (5′-3′) (Second amplification) |
|---|---|---|
| *cox1*-F | GTGGTGTTGATTTTTTGATGTTT | AGCAGGTGTTTCTAGAGTTTTTAGT |
| *cox1*-R | CCAAACGTAAACAACACTATAAAAGA | ACCCACCACAAAATAGGATCACT |
| *nad2*-F | GTTGAGCTATGTAATAATGTGTGGAT | GCGTTGATTCATTGATACATTG |
| *nad2*-R | AAATCTGTTGAATCTGCTACAACC | TAGTAAAGCTCAAACCGAGTTCT |
| *cob*-F | GGGTATGGCTTTGTATTATGGTAGTT | GTTTAAACTGGTAGATTGTGGTTC |
| *cob*-R | ATCACTCAGGCTTAATACTAACAGGAG | CTCCACAGTAGAAATCACCATCA |

were embedded into paraffin blocks and cut into 3-μm thick sections which were placed on glass slides. These sections were stained with H&E for pathological investigation using a Leica microscope. The characteristic features of *E. multilocularis* metacestode infection include the presence of PSCs, laminated and germinal layers, and host cell infiltration.

## DNA extraction

Genomic DNA was extracted from each sample of 75% ethanol-fixed liver using TIANGEN TIANamp Genomic DNA kits (TIANGEN BOITECH, Beijing, China) according to the manufacturer's instructions. DNA was eluted with 2×50 μl RNase free-water by centrifugation of the applied column at 12,000 rpm for 2 minutes, and the extracted DNA was stored at -20˚C until use. Nested-PCR and sequencing for identifying *E. multilocularis* and *Microtus* species were performed according to established methods [14–16]; primers used are listed in Table 1. In brief, 2 μl of each extracted DNA sample was added to a PCR tube containing 25 μl mixture comprising 2 primers and 21 μl H$_2$O. The reaction conditions comprised initial denaturation at 94˚C for 3 min, and then 35 cycles of 94˚C for 1 min, 56˚C for 1 min and 72˚C for 2 min with the last extension at 72˚C for 10 min. The PCR products were sent to BGI (Shenzhen, China) for sequencing. Bioedit (http://www.mbio.ncsu.edu/bioedit/bioedit.html) was used to compare the obtained sequences with sequences in GenBank.

## Statistical analysis

The Chi-Square test was used to compare the differences in AE prevalence/incidence over time and between counties and ethnic groups.

## Results

### High AE incidence from pasture areas in Yili Prefecture

The first AE case in our data set was identified in 1989 in Yili Friendship Hospital. By 2015, a total of 286 cases were diagnosed and registered in Yili Prefecture with an average annual incidence (AI) of 0.41/100,000 residents (Table 2). From 2005 to 2015, the AI was 0.70/100,000

**Table 2. The cumulative number of cases and the proportion of the population for the different ethnic groups in 2015.**

| Ethnic group | Cases | Cases proportion (%) | Population proportion (%) |
|---|---|---|---|
| Han | 54 | 18.88 | 30.36 |
| Kazak | 153 | 53.50 | 28.26 |
| Uygur | 35 | 12.24 | 23.28 |
| Hui/Dongxiang | 18 | 8.39 | 11.68 |
| Mongol | 24 | 6.29 | 1.23 |
| Xibo | 2 | 0.70 | 1.18 |
| Other ethnicities | 0 | 0 | 4.01 |

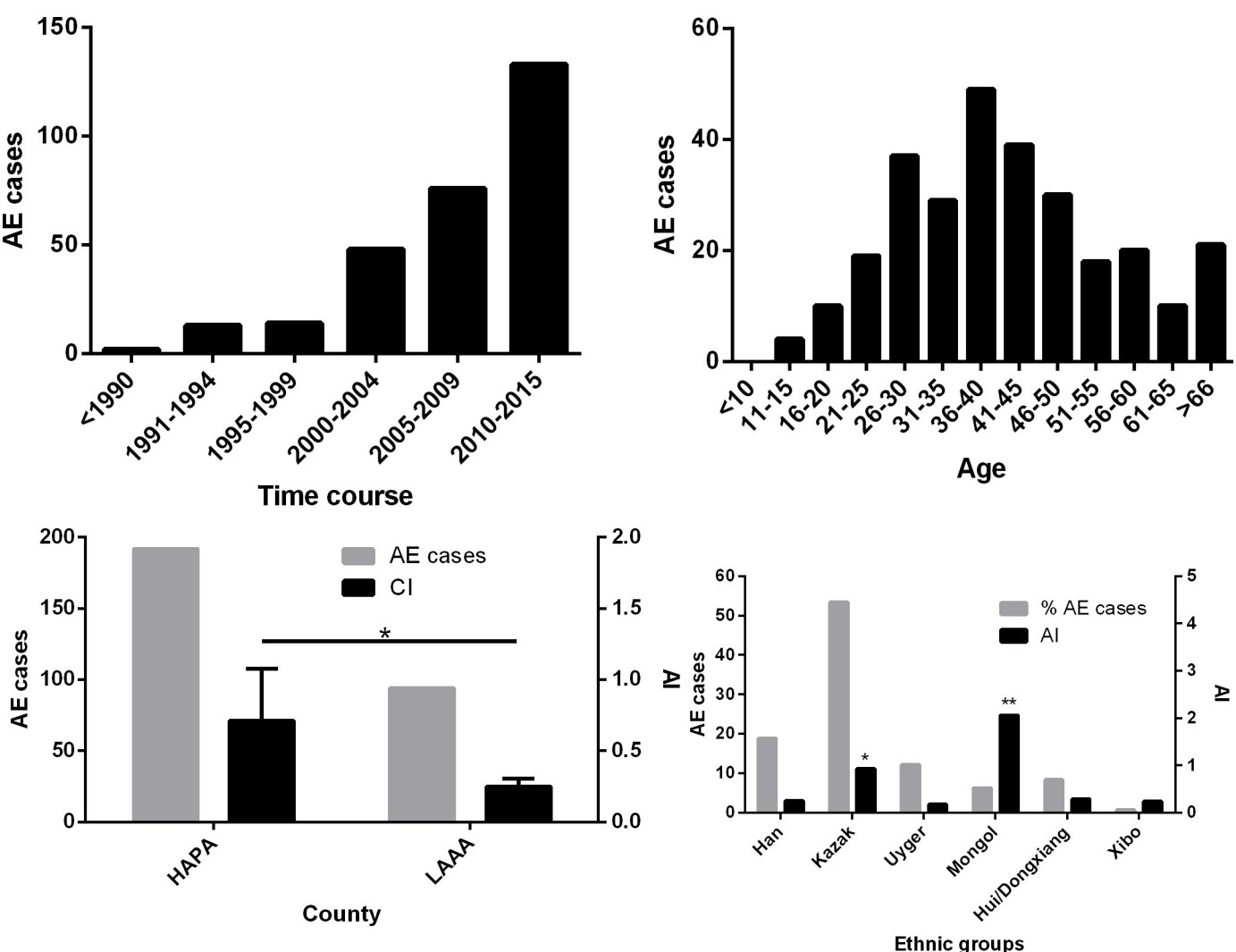

**Fig 2. Distribution of alveolar echinococcosis (AE) cases in Yili Prefecture, Xinjiang.** (a) Number of AE cases since 1989; (b) AE cases in different age groups; (c) AE cases and average annual incidence (AI, cases/100,000) between the high altitude mountainous pasture area (HAPA) and the low altitude mountainous agricultural area (LAAA); (d) AE cases and AI in different ethnic groups. An asterisk indicates a significant difference.

(209 cases/2,690,960 residents (2015)), which was 3.04 times of the AI in the first period from 1994 to 2004 (0.23/100,000). We found 73.08% (209/286) of these patients had been diagnosed from 2005 to 2015 (Fig 2). There was a significant difference in incidence between the two periods ($p<0.0001$). As shown in Fig 2, there is a linear relationship between incidence and time, with the incidence increasing over time ($\chi^2 = 4.403$, $p<0.05$). Of these AE patients, 153 cases were Kazak accounting for 53.50% of the total AE cases, followed by Han (18.88% (54/286)), Uygur (12.24% (35/286)), Hui/DongXiang (8.39% (24/286)), and Mongol (6.29% (18/286)) (Tables 2 and 3). Of the total cases, 150 were males and 136 females; more males than females were infected in all counties except Gongliu County, although the difference was not statistically significant (Table 4).

To estimate the risk and incidence of AE infection, we collected population data for Yili Prefecture from census and annual reports. According to the Sixth National Census in 2010

**Table 3. The distribution of AE patients in different ethnic groups.**

| County \ Ethnic | Han | Kazak | Uygur | Hui/Dongxiang | Mongol | Xibo | Total |
|---|---|---|---|---|---|---|---|
| Nileke | 12 | 34 | 3 | 3 | 8 | 0 | 60 |
| Xinyuan | 8 | 33 | 1 | 0 | 1 | 0 | 43 |
| Gongliu | 3 | 10 | 2 | 1 | 1 | 0 | 17 |
| Tekes | 4 | 16 | 0 | 5 | 0 | 0 | 25 |
| Zhaosu | 3 | 31 | 3 | 2 | 8 | 0 | 47 |
| Chabuchaer | 3 | 3 | 4 | 4 | 0 | 1 | 15 |
| Yining | 13 | 15 | 16 | 5 | 0 | 1 | 50 |
| Huocheng | 8 | 11 | 6 | 4 | 0 | 0 | 29 |
| **Total** | 54 | 153 | 35 | 24 | 18 | 2 | 286 |

(http://www.xjtj.gov.cn/), the population of Yili Prefecture was 2,397,551. A breakdown of the ethnic groups comprising this overall population in 2015, and the corresponding cumulative number of recorded cases for each group, are shown in Table 2.

To generate ethnic group specific AI rates, the total number of cases recorded for a given ethnic group was divided firstly by the total population of that group in 2015, then, divided by 26 (years), and multiplied by 100,000 to generate a AI measure of AE cases per 100,000 persons. This analysis was performed for each ethnicity and showed that the AI of AE in the Mongolian ethnic group was the highest (2.06/100,000 residents), 5.09 times higher than the average incidence in this area. The second highest risk group was Kazak (0.93/100,000), followed by Hui/Dongxiang (0.29/100,000), Han (0.25/100,000), Uygur (0.18/100,000) and Xibo (0.24/100,000) (Fig 2). Chi-Square analysis showed the differences in the AI between the Han and Kazak and Mongol ethnicities were highly significant ($p<0.0001$); there was no significant difference in the AI between the Han and Hui/Dongxiang, Uygur and Xibo ethnic groups.

Fig 2 shows that 87.76% of AE cases were young or middle-aged patients (between 16 and 60 years-old) with a peak age of 36–40. The mean and median ages for all AE cases were 41.86 and 46 years, respectively. All these cases were seen by hospital doctors, had pronounced lesion pathology, and were in poor health. Two cases had metastases that had spread into the lungs and two cases had brain metastases. Among all the AE cases, 71.43% had surgical resection immediately after diagnosis whereas the remainder did not undergo surgery due to medical reasons. All patients with AE were treated with albendazole. There were four young patients with one aged 12 (Hui/Dongxiang) and 3 (Kazakhs) aged 15. All were from sheep-farming

**Table 4. Sex distribution of AE patients in the study counties.**

| County \ Sex | Male | Female |
|---|---|---|
| Nileke | 35 | 25 |
| Xinyuan | 22 | 21 |
| Gongliu | 8 | 9 |
| Tekes | 16 | 9 |
| Zhaosu | 22 | 25 |
| Chabuchaer | 11 | 4 |
| Yining | 26 | 24 |
| Huocheng | 16 | 13 |
| **Total** | 156 | 130 |

families from HAPA, indicating that AE patients may have been infected with *E. multilocularis* at young ages, and that infection was likely associated with their family livelihood (Fig 2).

The AIs of four counties, Nileke, Zhaosu, Xinyuan and Tekes in the HAPA region were 1.22/100,000, 0.96/100,000, 0.51/100,000, and 0.55/100,000, respectively; these were significantly higher than the LAAA counties of Yining, Huocheng and Chabuchaer which had AIs of 0.19/100,000, 0.27/100,000 and 0.29/100,000, respectively ($p<0.05$) (Fig 2). The difference between Yining County and the four HAPA counties was highly significant ($p<0.0001$); however, there was no significant difference between Yining County and the other two LAAA counties ($p>0.05$).

### Follow-up of AE patients

Although contact with many of the patients was lost, contact was successfully made with 21 patients or their relatives. Of these cases, 14 were male and 7 were female, 15 were from the HAPA area with 6 from the LAAA area. By the time of being contacted, the youngest individual was 20 with the oldest being 81, with mean and median ages of 49.95 and 44 years respectively. Of these, 15 (15/21, 71.43%) had AE lesions surgically removed and were provided with albendazole treatment. However, 3 (3/15, 20%) of the patients relapsed and died within three years with an average survival time of 1.83 years after surgery. The majority (12/15, 80.00%) of the patients made good progress with an average survival time of 3.17 (2–5) years post-surgery prior to follow-up interviews. Six (28.57%) of the patients did not undergo surgery but were provided with albendazole. However, three died, having an average lifespan of 3.33 years after diagnosis; the median survival time of these 6 AE patients was 3.5 years. One patient had been taking albendazole for 3 years but was in poor health. The AE lesion of another patient was calcified after 5.5 years albendazole treatment and one patient remained in a stable condition for 3 years after taking the medication.

### High prevalence of *E. multilocularis* in small mammals from mountainous pasture areas

A total of 1411 small mammals were captured, of which 129 (9.14%) were infected with *E. multilocularis* metacestodes (Table 5), confirmed by both H&E staining and DNA sequencing.

**Table 5.  Alveolar echinococcosis in small mammals in Yili Prefecture, Xinjiang.**

| County / Species | Nileke | | Xinyuan | | Tekes | | Chabuchaer | | Total | |
|---|---|---|---|---|---|---|---|---|---|---|
| | Positive/ Samples | Infection prevalence | Positive/ Samples | Infection prevalence | Positive/ Samples | Infection prevalence | Positive/ Samples | Infection prevalence | Positive/ Samples | Infection prevalence |
| *Mus musculus* | 0 | 0 | 0 | 0 | 0 | 0 | 0/232 | 0 | 0/232 | 0 |
| *Microtus spp*[a] | 14/149 | 9.40% | 54/357 | 15.13% | 54/291 | 18.56% | 0/16[b] | 0 | 122/813 | 15.01% |
| *A. sylvaticus* | 0 | 0 | 0 | 0 | 0 | 0 | 1/249 | 0.40% | 1/249 | 0.40% |
| *R. sopimus* | 0 | 0 | 0 | 0 | 0 | 0 | 2/25 | 8.00% | 2/25 | 8.00% |
| *M. erythrourus* | 0 | 0 | 0 | 0 | 0 | 0 | 2/47 | 4.26% | 2/47 | 4.26% |
| *E. trancrei* | 2/2 | 0 | 0 | 0 | 0/2 | 0 | 0 | 0 | 2/4 | 50.00% |
| *Others*[*] | 0 | 0 | 0 | 0 | 0 | 0 | 0/41 | 0 | 0/41 | 0 |
| **Total** | 16/151 | 10.60% | 54/357 | 15.13% | 54/293 | 18.43% | 5/610 | 0.83% | 129/1411 | 9.14% |

Note: a, including *M. obcurus* and *M. arvailis*. b, these 16 *Microtus spp* were captured from the high-altitude mountainous pasture area.

*Others: *R. norvegicus*, (n = 29); *C. migratorius*, (n = 6); *M. tamariscinus*, (n = 6).

Through visual checking of 126 *Microtus spp* suspected liver samples, 122 were confirmed as *E. multilocularis*-positive by H&E staining with all the H&E positive samples subsequently further confirmed as having *E. multilocularis* infection by DNA sequencing. The 4 remaining livers were confirmed as negative by H&E staining, and the 20 non-infected control livers checked visually were all PCR-negative.

Among the small mammals collected, 813 were *Microtus* rodents captured in the HAPA including Xinyuan, Tekes and Nileke Counties in the up-stream areas of Yili River. These *E. multilocularis* positive animals accounted for 15.01% (122/813) of *Microtus spp* voles infected with *E. multilocularis* (Table 5). For identification of *Microtus* species, we PCR-amplified 270 *Microtus spp* DNA samples including 122 that were infected with *E. multilocularis* metacestodes. We obtained 159 *Microtus spp Cob* fragment sequences with the greatest number of sequenced samples originating from *E. multilocularis* infected animals (accession numbers: MN049933-MN049940, MN049943-MN049947, MN049950-MN049954). The DNA sequence analysis showed that there were two species of *Microtus spp* present in the study location: 96.23% (153/159) were *M. obscurus* and 3.77% (6/159) were *M. arvalis;* 33.59% (52/153) and 33.33% (2/6 of *M. obscurus* and *M. arvalis* were, respectively, infected with *E. multilocularis*. However, only two *Microtus spp* voles harboured *E. multilocularis* metacestodes with PSCs present. Four *Ellobius tancrei* were captured with two (2/4) infected with metacestodes, both with PSCs of *E. multilocularis* (Fig 3). In Chabuchaer County in the LAAA 594 small mammals were captured; *Apodemus sylvaticus* (249/594) and *Mus musculus* (232/594) were predominant. Only one *A. sylvaticus* (0.40%) was infected with *E. multilocularis* (Table 5). We found that 2 of 47 *Meriones erythrourus*, and 2 of 25 *Rhombomys opimus* were infected with *E. multilocularis* metacestodes (Table 5). No *Mus musculus* (n = 232), *Rattus norvegicus* (n = 29), *Cricetulus migratorius* (n = 6) or *M. tamariscinus* (n = 6) (Table 5) were infected with *E. multilocularis* metacestodes

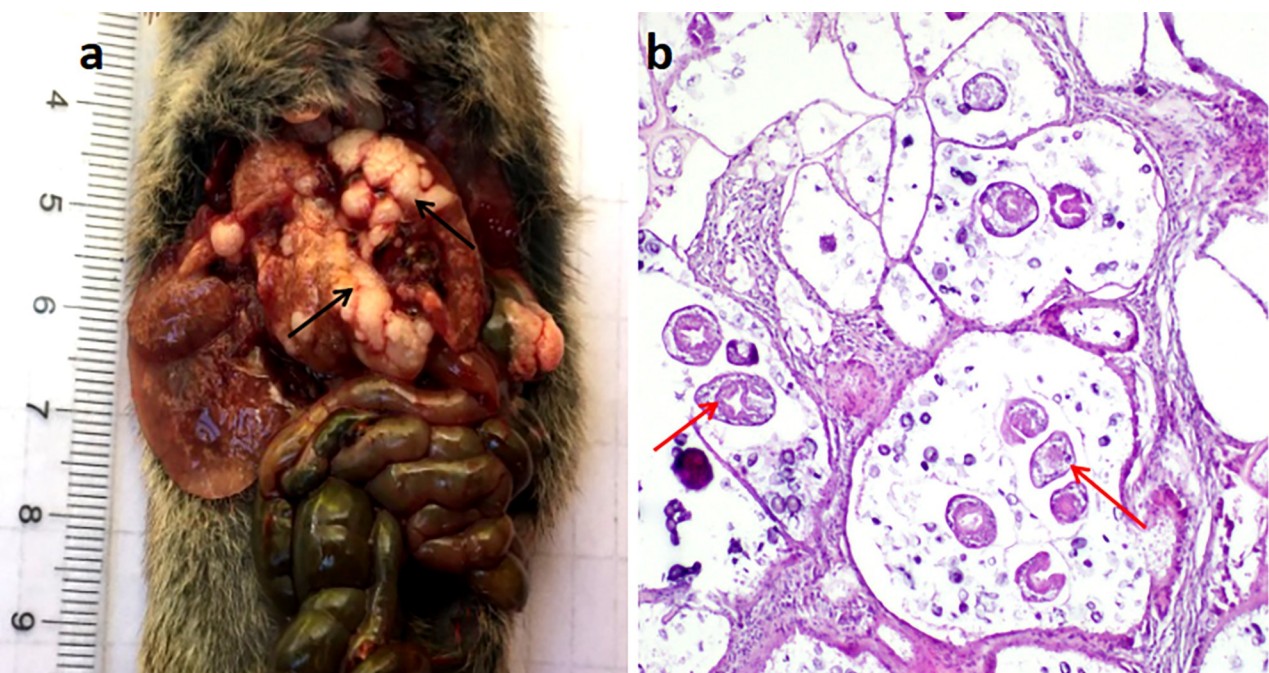

**Fig 3. Haematoxylin and eosin staining showing the characteristic pathological response in alveolar echinococcosis.** (a) *E. multilocularis* metacestode; (b) Liver section with PSCs, 200× original magnification. Black arrow indicates cystic lesions; red arrow indicates PSCs.

### *E. multilocularis* genotypes

To further confirm the identity of the 129 H&E-positive samples, PCR was used to amplify three gene (*cox*1, *nad*2 and *cob*) fragments of *E. multilocularis* and then the PCR products were directly sequenced to determine genetic variation among the isolates. A total of 66 isolates produced one or three gene sequences for analysis. The sizes of the amplified DNA fragments were 497 bp, 346 bp and 397 bp for *cox*1, *nad*2, and *cob*, respectively (GenBank accession numbers: *cox*1:MH211144-MH211159, *nad*2: MH211160-MH211174 (except to MH211170), *cob*: MH211175-MH211190).

The DNA sequence analysis showed that the *E. multilocularis cox1*, *nad2* and *cob* sequences of the isolates from Yili Prefecture were genetically similar to isolates from Central Asia and Europe, indicating a close evolutionary relationship. We amplified 43 copies of the *cox1* gene and divided them into 16 haploids (Fig 4A), with XH1 the most common in 21 samples. Distance-based NJ analysis of the *cox1* gene sequences showed XH2, XH3, XH13, XH16 were close to the published sequences KT965441 (Ningxia, China) and KT965439 (Xinjiang, China)

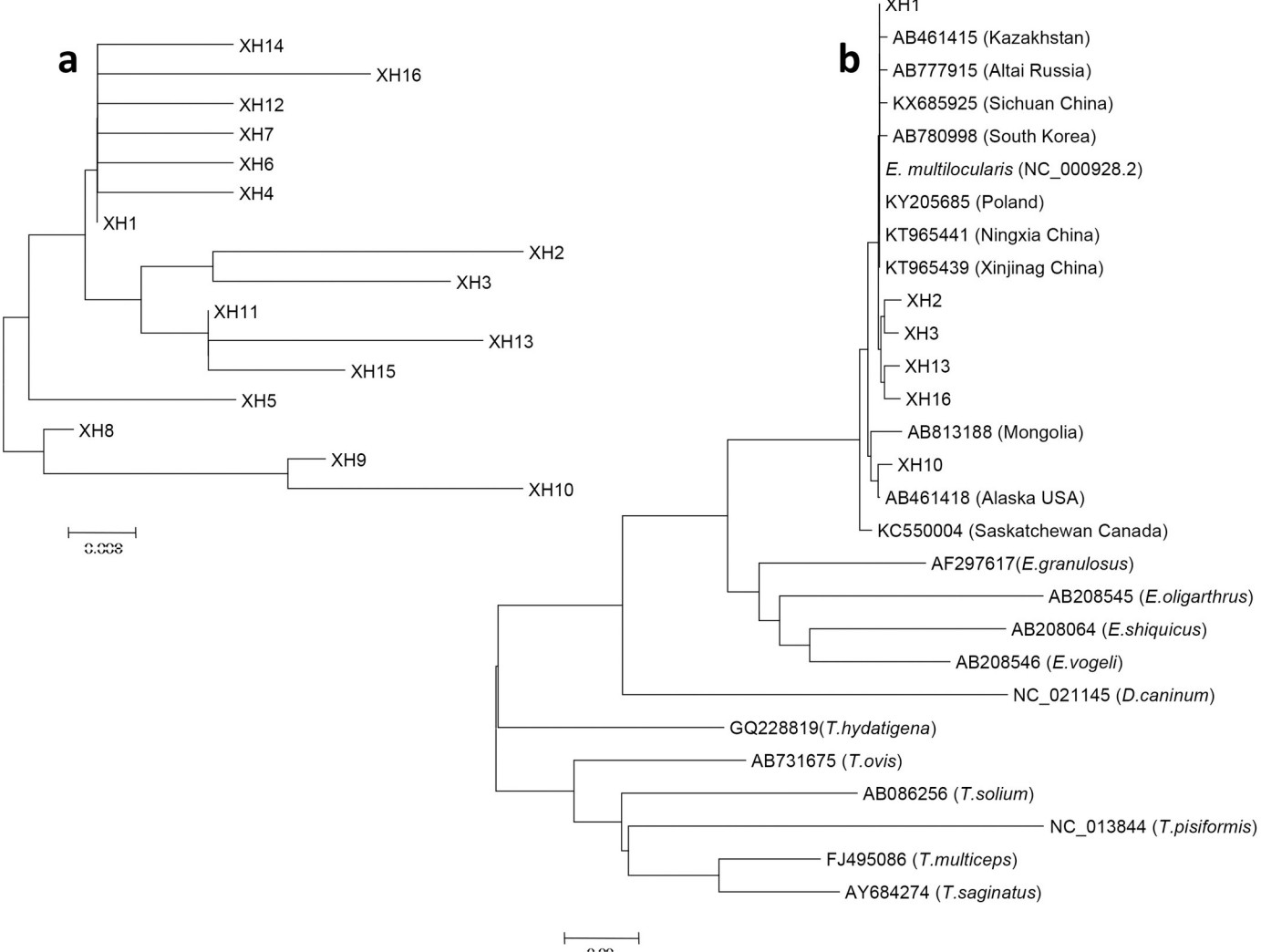

**Fig 4. Phylogenetic tree of *E. multilocularis* mtDNA *cox1* gene.** The phylogenetic tree was constructed using the neighbor-joining algorithm of the phylogeny program MEGA 6.0. Bootstrap method via 1000 pseudo replicates was used to assess the reliability of the tree. (a) Sixteen haplotypes identified in this study; (b) Sixteen haplotypes were analyzed with other similar sequences in the phylogenetic tree. XH indicates the *cox1* gene haplotype.

**Table 6. *E. multilocularis* haplotypes characterized by partial *cox1* sequences used for phylogenetic analysis.** All samples were from small mammal hosts collected during the current study.

| Haplotype | Location | Number | Accession number | Host |
|---|---|---|---|---|
| XH1 | Xinyuan (10), Tekes (5), Nileke (4), Chabuchaer (2) | 21 | MH211144 | *Microtus spp, E. trancrei* |
| XH 2 | Nileke (1) | 1 | MH211145 | *Microtus spp* |
| XH 3 | Tekes (1) | 1 | MH211146 | *Microtus spp* |
| XH 4 | Chabuchaer (1) | 1 | MH211147 | *Microtus spp* |
| XH 5 | Xinyuan (1) | 1 | MH211148 | *Microtus spp* |
| XH 6 | Tekes (1) | 1 | MH211149 | *Microtus spp* |
| XH 7 | Xinyuan (1) | 1 | MH211150 | *Microtus spp* |
| XH 8 | Tekes (1) | 1 | MH211151 | *Microtus spp* |
| XH 9 | Xinyuan (4), Tekes (2) | 6 | MH211152 | *Microtus spp* |
| XH 10 | Xinyuan (1) | 1 | MH211153 | *Microtus spp* |
| XH 11 | Nileke (1) | 1 | MH211154 | *Microtus spp* |
| XH 12 | Xinyuan (1) | 1 | MH211155 | *Microtus spp* |
| XH 13 | Xinyuan (1) | 1 | MH211156 | *Microtus spp* |
| XH 14 | Tekes (1) | 1 | MH211157 | *Microtus spp* |
| XH 15 | Xinyuan (2), Chabuchaer (1) | 3 | MH211158 | *Microtus spp* |
| XH 16 | Xinyuan (1) | 1 | MH211159 | *Microtus spp* |

(Fig 4B). We found that the XH1 sequence was 100% identical to the haplotype reported from Kazakhstan and China (Sichuan Province) [17–18]. The other haplotypes detected in 16 isolates in the present study were identical to sequences previously published in GenBank (Table 6). The one predominant (60/97) *E. multilocularis Nad*2 haplotype (MH211160) present in the small mammals was identical to the sequence isolated from local human AE cases in a previous study of ours [14] (S1 Fig).

## Discussion

### Yili Prefecture is a high endemic area for AE

In this study, we retrospectively collected human AE cases from eight counties and one city in Yili Prefecture, Xinjiang Autonomous Region. The results showed that AE incidence increased over a 20 year period, especially in the last decade, and that the incidence of AE is likely associated with the presence of altitude-associated small mammal species such as *Microtus* which are the predominant rodents in the pasture areas. In a preliminary study, Zhou et al.[10] reviewed human AE cases in Yili Prefecture and showed that the Yili valley area was an endemic area with a incidence of 3.80 cases/100,000 inhabitants (n = 77 cases), with 10 cases recorded before 1990 [10]. Here, all the AE cases in 4 hospitals were treated by approved teams with specific expertise for surgical treatment of AE; these included 3 hospitals in Yili Prefecture and one in Urumqi (First Affiliated Hospital of Xinjiang Medical University), with records showing the earliest AE case was registered in 1989. We found that a total of 286 AE patients were diagnosed and treated in Yili Prefecture between 1989 and 2015, with more than 200 cases recorded after 2000. It is difficult to obtain true incidence figures by sampling methods in the Yili Prefecture area as the semi-nomadic communities drive their livestock seasonally between summer and winter pastures. Also, given the relatively low abundance of AE, establishing true incidence values requires a large population to be surveyed which is logistically difficult to perform in such remote communities. China commenced control of echinococcosis program in 2005. Cumulative incidence (CI) has been used to compare the endemic picture for AE over two time periods [12] but in this study, we used annual incidence (AI) to compare the endemic situation over two periods of time, i.e. 11

years before commencement of the control program and 11 years after commencement of the control program. The AI more than doubled in the second survey period (2005–2015) compared with the first (1994–2004) with 0.70 cases per 100,000 residents versus 0.23 per 100,000, indicating that Yili Prefecture is both a highly endemic and an emerging area for AE [8,19–21]. In the last 10 years, the extensive education program and improvement of diagnostics capability may have impacted on the number of recorded AE cases. However, data from 2011–2015 showed that 20, 20, 33 (a partial mass ultrasound survey was undertaken in 2013), 18 and 17 cases respectively were identified, indicating a levelling trend of AE cases.

Using the AI, this study also showed that the AE incidence in the Kazak population (0.93/ 100,000) was, respectively, 3.70, 5.31, 3.20 and 3.89 times higher than that in Han, Uygur, Hui/ Dongxiang and Xibo groups in Yili Prefecture. Although the overall number of cases in the Mongolian population was low, the AI (2.06/100,000) was the highest of all the ethnic groups, being 8.19, 11.76, 7.09 and 8.62 times higher than the corresponding incidence values for the Han, Uygur, Hui /Dongxiang and Xibo ethnic groups, respectively.

The reasons for the Kazak and Mongol ethnicities presenting with a higher incidence may be due to these groups mainly working with animal production following traditional methods, including moving seasonally in search of pasture for their animal herds [10]. Living in the high-altitude mountain pasture areas from May to the end of September is a major reason why human infection is more likely due to the high numbers of *Microtus spp* and *Ellobius* voles infected with *E. multilocularis*. In these HAPA, sheep dogs rely, at least partly, on feeding on rodents for food, and the high parasite prevalence in these animals likely results in a significant numbers of dogs infected with *E. multilocularis*. One limitation of the current study is that no prevalence data were available on dog infections with adult *E. multilocularis* due to the strict bio-safety restrictions operating at the First Affiliated Hospital of Xinjiang Medical University. However, the families of sheep-farmers generally raise one or more sheepdogs at any particular time, and these dogs likely play an important role in the transmission of AE to humans.

Age group analysis showed the peak age group for human *E. multilocularis* infection was 36 to 40 years, with 77.27% of the AE patients being in the broad age range of 21–55. There is normally a 5–15 year period between initial infection and patients beginning to exhibit symptoms with AE [22]; therefore it is likely that AE patients aged 30–40 became infected when they were 15–30 years old. In this study, the youngest patient was 11 years old, suggesting the child had been infected early on in life.

## Treatment and follow-up

Most of the AE patients revealed in this study only visited hospitals and were diagnosed by clinicians in the later stages of the disease. For these AE patients, surgically removing lesions is the first option for treatment. Although we could not follow-up all the patients, more than half were in a stable condition following surgery and albendazole treatment. One patient had stopped taking albendazole. For those patients who did not undergo surgery, more than half died within 1–3 years, even when receiving albendazole treatment. It is reported long-term treatment with albendazole has improved the 10-year survival rate of AE patients compared with untreated historical controls from 6–25% to 80–83% [23–27]. It has also been suggested that the majority of AE cases can recover after surgery [2], while the majority of patients not having surgery die or have a poor prognosis.

## Yili Prefecture is a natural focus of AE

This study confirms Yili Prefecture as a natural focus of AE and demonstrates that *Microtus spp* and *Ellobius* voles harbor high levels of infection with *E. multilocularis*. Small mammal

species include *M. obscurus* and *M. arvalis*, *E. trancrei* in the HAPA, and *Mus musculus*, *R. norvegicus*, *A. sylvaticus*, *M. erythrourus* and *R. opimus* in the LAAA. However, *Microtus spp* were by far the dominant rodents captured, accounting for 99.50% of all small mammals caught in the HAPA.

In the LAAA areas of the Yili River valley region, *A. sylvaticus* and *M. musculus* were dominant, accounting for 39.06% (232/594) and 49.49% (249/594) of the rodent species present, respectively. However, no *M. musculus* or *R. norvegicus* (brown rats) were found infected with *E. multilocularis* metacestodes, and only one *A. sylvaticus* was shown infected. Accordingly, although *M. musculus*, brown rats and *A. sylvaticus* were dominant rodent species in the LAAA, *M. erythrourus* and *R. opimus* may play major roles in the transmission of *E. multilocularis* in this area.

The prevalence of *E. multilocularis* in *Microtus spp* was 15.01%. All these voles were captured from the HAPA between 1980 and 2500 meters above sea level. Geographically, altitude and rainfall impact the landscape and habitats, with grasslands favorable to *Microtus spp* voles. This may in turn impact the transmission of AE [28–35]. In Yili Prefecture, four counties in the higher altitude areas had higher AI of human AE than those counties in lower altitude areas. In the high mountainous areas, *Microtus spp* voles were the dominant small mammals captured and these had a very high prevalence of *E. multilocularis*. The distribution of human AE seems to be associated with the distribution of abundant vole populations [34–36]. We identified 0.40% of *A. sylvaticus* infected with *E. multilocularis* metacestodes. In addition, in the low altitude mountainous areas, 8.00% of *R. opimus* and 4.26% of *M. erythrourus* were infected, indicating that these two species may play a role in the transmission of AE although they are not the dominant species present [33,37].

Among the other small mammal species captured, four *E. trancrei* were recorded, with two infected. This species is more subterranean than *Microtus spp.* and is widespread on the grasslands of Central Asia [32]. Its role in *E. multilocularis* transmission needs clarification as it depends on the still unknown capacity of dogs and fox to prey on this species [38,39].

*E. multilocularis* is common on the Qinghai-Tibetan Plateau [7,8,40,41] and in Central Asia [20,42]. Geographically these two endemic areas are separated by the Kunlun and Tianshan Mountains and the Taklamakan Desert. In this study, we determined that DNA sequences of *E. multilocularis* isolates from Yili were genetically similar to isolates from Central Asia and Europe [43–46].

Here, small mammals were captured using a "drowning method" leading to their sampling close to nearby water points. In the summer season, herdsmen are highly dispersed in the mountain pastures and live in tents generally situated 10–50 meters away from streams or creaks. Our field survey showed that in these areas the density of voles was high, resulting in an increased risk of infection in dogs accompanying the herdsman due to their being in close proximity to higher densities of *E. multilocularis* intermediate hosts. In general, the sustainability of transmission of *E. multilocularis* depends on a wildlife, predator/prey, cycle mainly involving foxes/dogs and 40 species of small mammals with transmission impacted by landscape changes [33,34,37]. We found that there were only two species of small mammals from the HAPA, and seven species from the LAAA. The results indicate that the prevalence of AE was higher in areas of relatively low small mammal biodiversity than in more diverse host communities. In low diversity small mammal communities, potential intermediate host species are prone to population outbreaks. Also, grass productivity and forest cover are associated with rainfall and temperature, and these features are correlated (locally) with altitude [32]. Consequently, this resulted in an increased risk of infection in herdsman. Dosing dogs at least once a month with baited praziquantel may prevent the transmission of *E. multilocularis* to this herdsmen community [47].

## Conclusion

In summary, Yili Prefecture, Xinjiang is highly endemic for AE where voles (*Microtus spp.*) likely play a crucial role in its transmission. The high AE incidence in the Kazak and Mongolian communities is probably due to farmers raising sheep in the summer season in high-altitude pasture areas where there are significant numbers of small mammals infected with *E. multilocularis;* this enhances AE transmission due to close contact of these farmers with their sheep dogs.

## Supporting information

**S1 Fig. Identical *nad2* sequences for *E. multilocularis* samples isolated from small mammals and local human AE cases reported in a previous study of ours [14].**
(TIF)

## Acknowledgments

We are grateful to the Centers for Disease Control and Prevention for their assistance in the capture and autopsy of rodents in Xinyuan, Nileke and Zhaosu. We also thank Mr. Ma Jianjun for providing information about the rodent sampling locations.

## Author Contributions

**Conceptualization:** Patrick Giraudoux, Wenbao Zhang, Jun Li.

**Data curation:** Baoping Guo, Gang Guo, Haiyan Wang, Jianjun Ma, Ronggui Chen, Xueting Zheng, Jianling Bao, Li He, Tian Wang, Wenjing Qi, Mengxiao Tian, Junwei Wang, Canlin Zhou, Patrick Giraudoux, Christopher G. Marston, Donald P. McManus, Wenbao Zhang, Jun Li.

**Formal analysis:** Baoping Guo, Patrick Giraudoux, Christopher G. Marston, Donald P. McManus, Wenbao Zhang, Jun Li.

**Funding acquisition:** Wenbao Zhang, Jun Li.

**Investigation:** Baoping Guo, Zhuangzhi Zhang, Gang Guo, Haiyan Wang, Jianjun Ma, Ronggui Chen, Xueting Zheng, Jianling Bao, Li He, Tian Wang, Wenjing Qi, Mengxiao Tian, Junwei Wang, Canlin Zhou, Wenbao Zhang, Jun Li.

**Methodology:** Baoping Guo, Patrick Giraudoux, Christopher G. Marston, Wenbao Zhang, Jun Li.

**Writing – original draft:** Baoping Guo, Zhuangzhi Zhang, Yongzhong Guo, Wenbao Zhang, Jun Li.

**Writing – review & editing:** Patrick Giraudoux, Christopher G. Marston, Donald P. McManus, Wenbao Zhang, Jun Li.

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
