## [Decision Letter · Decision Letter 0]

29 Mar 2020

Dear Dr. Zhang,

Thank you very much for submitting your manuscript "High endemicity of alveolar echinococcosis in Yili Prefecture, Xingjiang, China is associated with pasture landscape and the distribution of Microtus spp. and animal farming activity" for consideration at PLOS Neglected Tropical Diseases. As with all papers reviewed by the journal, your manuscript was reviewed by members of the editorial board and by several independent reviewers. In light of the reviews (below this email), we would like to invite the resubmission of a significantly-revised version that takes into account the reviewers' comments. 

There are some substantial issues, particularly those identified by the second reviewer. The manuscript can be reconsidered only if all these issues are adequately addressed.

We cannot make any decision about publication until we have seen the revised manuscript and your response to the reviewers' comments. Your revised manuscript is also likely to be sent to reviewers for further evaluation.

Sincerely,

Paul Robert Torgerson

Guest Editor

Mar Siles-Lucas

Deputy Editor

There are some substantial issues, particularly those identified by the second reviewer. The manuscript can be reconsidered only if all these issues are adequately addressed.

Reviewer's Responses to Questions

**Key Review Criteria Required for Acceptance?**

**Methods**

-Are the objectives of the study clearly articulated with a clear testable hypothesis stated?

-Is the study design appropriate to address the stated objectives?

-Is the population clearly described and appropriate for the hypothesis being tested?

-Is the sample size sufficient to ensure adequate power to address the hypothesis being tested?

-Were correct statistical analysis used to support conclusions?

-Are there concerns about ethical or regulatory requirements being met?

Reviewer #1: Please see below.

Reviewer #2: Information needs to be provided on when (year/season) small mammal trapping occurred. 

The human aspect of the paper largely follows the methods of Zhou et al. (2000). That being said, additional information is needed in the Methods on who these AE patients represent (there is some additional information provided in the Discussion). The cases appear to be those with severe clinical disease seen in facilities with surgical capabilities. Are these the only hospitals/clinics in the region that treat AE patients? Due to the chronic nature of AE and the presumed lack of medical access for some cases, a clear description of the study population needs to be supplied so that the reader can understand the measure(s) of frequency used.

Even though case numbers are small, displaying the values as cumulative prevalence results in lost information. The authors are also not consistent with their frequency of infection terminology (e.g., switching between prevalence and incidence).

Realizing the difficulty in reconnecting with patients in this location, patient follow-up information is largely incomplete. Any form of follow-up contact could only be made with 7.3% of patients (n=21) so it is difficult to draw any conclusions from this information. In addition, no information is provided about these patients’ year of treatment, age, sex, or comorbidities. 

Data analysis methods are very simple and don’t make any attempt to evaluate associations between human AE cases and small mammal distribution, landscape, or farming activity as indicated by the paper’s title.

**Results**

-Does the analysis presented match the analysis plan?

-Are the results clearly and completely presented?

-Are the figures (Tables, Images) of sufficient quality for clarity?

Reviewer #1: Please see below.

Reviewer #2: Much of the results can be summarized in tables rather than presented in the text.

The abstract states that "...E. multilocularis DNA sequences from small mammals were identical to isolates from local AE cases." However, I don't see where sequence data from local human cases were obtained or discussed as part of this study. Please clarify. 

Please confirm the geographic origins of sequences KT965441 and KT965439. They differ between the text and figure 4.

**Conclusions**

-Are the conclusions supported by the data presented?

-Are the limitations of analysis clearly described?

-Do the authors discuss how these data can be helpful to advance our understanding of the topic under study?

-Is public health relevance addressed?

Reviewer #1: Please see below.

Reviewer #2: There is quite a bit of repetition of Results in the Discussion.

It would be helpful if the flow of the Discussion mirrored the Methods and Results rather than moving from human AE frequency, to small mammal distribution, back to clinical management of AE cases.

**Editorial and Data Presentation Modifications?**

Reviewer #1: (No Response)

Reviewer #2: The authors refer to the use of experimental animals (line 100). Please clarify if this is simply another reference to the trapped small mammals or if there was a separate group of animal subjects.

**Summary and General Comments**

Reviewer #1: High endemicity of alveolar echinococcosis in Yili Prefecture, Xingjiang, China is associated with pasture landscape and the distribution of Microtus spp. and animal farming activity

The study demonstrated the status of Alveolar Echinococcosis in Yili prefecture in Xinjiang region of China based on historic hospital and field data in the last two decades. The area is one of the most endemic regions of AE in the world and the information provided in the study are required for implementation of any control program. Several issues need to be addressed before the manuscript be considered for publication:

Major points

- Line 132 & 200: "To calculate the prevalence of AE" and "to estimate the annual prevalence over the two decades". As the authors investigated the occurrence of new AE cases during a period of time, it is more accurate to use "Incidence".

- Line 168-176: No need to provide much details on DNA extraction method, "according to the manufacturer’s instructions" is fine.

- No data have been provided for sex distribution of the patients. Please tabulate data on the age- and sex-specific incidence of AE in different ethnicities/regions.

- Please provide the Mean and Median age of the patients.

- Patient survival data are very valuable. It would be great to tabulate survival data and provide 5-year survival rates (absolute or relative) of AE patients (the percentage of AE patients living five years after the disease is diagnosed).

Also please report the Median survival that is the period of time after which 50% of the patients have died and 50% have survived.

- Were there any patients undergoing liver transplantation? If yes please provide details of the prognosis and patients conditions.

- The method used for detecting protoscoleces in the cysts is not clear.

- Line 262: "We also captured 16 from HAPA in Chabuchaer County", this county has already been categorized in LAAA region! Please clarify.

- Please provide chi-square analysis results for comparing differences among time periods and ethnic groups.

- Contradictory statements written in line 280-283 on Rhombomys and Meriones infection ! 

- In Fig. 1 no sampling site (cross) has been specified for the farming areas. Does it mean all the samples were taken from the pasture areas?

- According to the Fig. 2, several AE cases were observed in adolescents under 15 years of age. this is an interesting finding. Please provide additional data on these AE patients. The nature of AE in children and adolescents is unknown therefore these type of information are very valuable. 

- It is not clear which haplotype from where? has the highest frequency. To understand the extent of genetic variation within E. multilocularis isolates from Xinjiang please provide (in a table) more data on the frequency distribution of 16 haplotypes according to different locations and other variables if applicable. 

- Line 204: The study found that 73.08% of AE patients had been diagnosed in the last 10 years, please specify that if this is because of more diagnostic facilities provided in recent years or more active case-finding or more intense disease transmission or more patients seeking treatment etc.? please elaborate more on this.

- line 382: "Microtus suppress the production of PSCs." As the method used for PSC detection is not clear, this statement is not justifiable. Several studies across the continents have shown that different Microtus spp. are highly potent intermediate hosts for E. multilocularis, i.e they are favorite preys for the definitive hosts, susceptible to E. multilocularis with a high population density (see Ecology and Life Cycle Patterns of Echinococcus Species, Adv. Parasitol. 2017; and Giraudoux et al. 2013). Therefore as no evidence has been presented in the study, it is quite premature to say "Microtus spp. suppress protoscoleces production". In addition the authors demonstrate further contradictions in the Title (line 1-3) and Conclusion section (line 414-415) when they talked about the crucial role of Microtus in AE transmission in the region ! Please elaborate more on this.

- Please provide separate descriptions for the two phylogenetic trees in the legend for Fig. 4. 

Minor points

- line 177: PCR or nested-PCR ?

- Line 206: "Kazak (53.50% (153/286))" is redundant.

- line 280: "2 of 25 Rhombomys opimus," is redundant.

- Line 297: the GenBank accession number KT965411 is for Influenza A virus !

- Please be consistent with reporting decimals, choose either one or two decimal places all over the manuscript.

- line 365: Em > E. multilocularis

Thank you.

Reviewer #2: The title does not appear to align with the study’s contents. No formal spatial or statistical evaluation was conducted between human AE cases and small mammal distribution, landscape, or farming activity. 

The study is comprised of two parts. The first describes AE cases from Yili Prefecture seen at four regional hospitals from 1989-2015.The second part describes the proportion of trapped small mammals from four counties in the same prefecture that were positive for AE. Parasite sequence analysis was subsequently performed on samples obtained from these small mammal intermediate hosts. There are no true study objectives provided and, unlike the title would suggest, only basic statistical comparisons were performed.

PLOS authors have the option to publish the peer review history of their article (what does this mean?). If published, this will include your full peer review and any attached files.

Reviewer #1: Yes: Dr. Majid Fasihi Harandi

Reviewer #2: No
---

## [Decision Letter · Decision Letter 1]

15 Jun 2020

Dear Dr. Zhang,

Thank you very much for submitting your manuscript "High endemicity of alveolar echinococcosis in Yili Prefecture, Xingjiang, China: the prevalence of the disease in different ethnic communities and small mammals" for consideration at PLOS Neglected Tropical Diseases. As with all papers reviewed by the journal, your manuscript was reviewed by members of the editorial board and by several independent reviewers. In light of the reviews (below this email), we would like to invite the resubmission of a significantly-revised version that takes into account the reviewers' comments. 

We cannot make any decision about publication until we have seen the revised manuscript and your response to the reviewers' comments. Your revised manuscript is also likely to be sent to reviewers for further evaluation.

Sincerely,

Paul Robert Torgerson

Guest Editor

Mar Siles-Lucas

Deputy Editor

Reviewer's Responses to Questions

**Key Review Criteria Required for Acceptance?**

**Methods**

-Are the objectives of the study clearly articulated with a clear testable hypothesis stated?

-Is the study design appropriate to address the stated objectives?

-Is the population clearly described and appropriate for the hypothesis being tested?

-Is the sample size sufficient to ensure adequate power to address the hypothesis being tested?

-Were correct statistical analysis used to support conclusions?

-Are there concerns about ethical or regulatory requirements being met?

Reviewer #1: Please see "Summary and General Comments".

Reviewer #2: See summary and general comments.

**Results**

-Does the analysis presented match the analysis plan?

-Are the results clearly and completely presented?

-Are the figures (Tables, Images) of sufficient quality for clarity?

Reviewer #1: Please see "Summary and General Comments".

Reviewer #2: See summary and general comments.

**Conclusions**

-Are the conclusions supported by the data presented?

-Are the limitations of analysis clearly described?

-Do the authors discuss how these data can be helpful to advance our understanding of the topic under study?

-Is public health relevance addressed?

Reviewer #1: Please see "Summary and General Comments".

Reviewer #2: See summary and general comments.

**Editorial and Data Presentation Modifications?**

Reviewer #1: Please see "Summary and General Comments".

Reviewer #2: See summary and general comments.

**Summary and General Comments**

Reviewer #1: Following most of the reviewers comments, the manuscript is remarkably improved. However several comments have not been addressed in the revised version:

- Ref to the Reviewer #2 comment, in Discussion, the "Treatment and follow-up" section has come after small mammal section, please follow the reviewer comment and put all clinical human data together.

- Line 168: "RNA free-water" is not correct.

- "there is no need to do chi-square test" So please modify the Statistical analysis section in M&M (Ref to the Reviewer #1 comment).

- Ref to the Reviewer #1 comment, please note that Em, Eg, Ro, As, Et are taxonomically / biologically non-sense. Please use standard scientific nomenclature. 

- English writing needs a revision.

- line 442: was preformed > was performed

Reviewer #2: The manuscript is somewhat improved. However, numerous grammatical errors still exist. In addition, the authors make several statements regarding relationships between human AE cases and perceived risk factors that are not directly supported by study data. Please find my specific comments below (line numbers refer to the Word document with track changes). 

Based on the study design, I would suggest that the authors moderate the statement, “The overall results indicate that the transmission of AE is highly associated with landscape and distribution of M. obscurus in Yili Prefecture” since they didn’t evaluate landscape or statistical associations between small mammal distribution and human AE cases. (lines 50-52) [Wording in the author summary is slightly better.]

I’m also not sure how the authors came to the conclusion that “…sheep-farming activity is a risk for infection”. (line 43)

Information in Table 6 (E. multilocularis haplotypes and accession numbers) needs to be elaborated upon. It is not clear if these samples were from small mammals or humans (or if these are samples from the current study or a previous study). The authors also need to provide support for the statement, “…E. multilocularis DNA sequences from small mammals were identical to isolates from local human AE cases”. (lines 49-50) 

The statement in lines 93-96 (“In this study, we retrospectively collected human AE cases from eight counties and one city in Yili Prefecture, which shows that AE prevalence was increasing in the 20 years period, especially in the last decade and the prevalence of AE is likely associated with landscape features.”) doesn’t belong in the introduction. Again, this study did not evaluate landscape features.

The authors state that, “The Chi-Square test was used to compare the differences in prevalence between time periods, counties, and ethnic groups.” However, I don’t see where chi-square values (and associated p-values) for time periods are presented in the Results section. In addition, chi-square findings for ethnic groups and counties are not clearly presented (please also see my comment regarding the figure 2 legend). (lines 199-200) 

Are the provided ages for the follow-up patients, their ages at the time of follow-up or at diagnosis? What about those who had died during the follow-up period? In general, the “follow-up of AE patients” section is very difficult to follow and would benefit from restructuring and clarification when presenting information about time since diagnosis and time since surgery for both patients who are still alive and for those who have died.

Please clarify what is meant by “21 samples working”. (line 343)

Figure 2 legend- What do the authors mean by “an asterisk indicates above average”? 

Figure 3 doesn’t appear to be referenced in the text.

The authors added the following section to the revised document: “An ultrasound survey about 4000 residents per county on echinococcosis was preformed in 2013(data not shown). In this year, a total of 33 AE cases were registered, which is 11.40 more than the average (21.60) of AE cases from 2012 to 2016 (with AE cases being 20, 33, 20, 18 and 17 respectively), indicating that the mass screen had a limit impact on the growth of AE cases in the last 10 years.” This statement is confusing and I don’t think it addresses the question regarding possible changes to patient access to care over the study time period. 

Overall, the flow of the Discussion remains disjointed.

PLOS authors have the option to publish the peer review history of their article (what does this mean?). If published, this will include your full peer review and any attached files.

Reviewer #1: Yes: Majid Fasihi Harandi

Reviewer #2: No
---

## [Decision Letter · Decision Letter 2]

31 Aug 2020

Dear Dr. Zhang,

Thank you very much for submitting your manuscript "High endemicity of alveolar echinococcosis in Yili Prefecture, Xingjiang Autonomous Region, the People’s Republic of China: the prevalence of the disease in different ethnic communities and in small mammals" for consideration at PLOS Neglected Tropical Diseases. As with all papers reviewed by the journal, your manuscript was reviewed by members of the editorial board and by several independent reviewers. The reviewers appreciated the attention to an important topic. Based on the reviews, we are likely to accept this manuscript for publication, providing that you modify the manuscript according to the review recommendations. 

The reviewers have found the revision largely satisfactory. However, before final acceptance of this manuscript, the authors need to ensure that reporting of population data with regard to diseases is standardized and using the correct epidemiological terms. For human cases is incidence per 100,000 per year. Please ensure all human incidence data is reported as such. From the materials and methods it seems the “cumulative prevalence” was based on the period 1989-2015, thus over a period of 27 years. This is confusing when making comparisons to other studies and gives a false impression that the incidence is somewhat higher than it is in reality.

In the author summary the way prevalence is reported is just wrong. Prevalence is the proportion with the disease at one point in time...ie if an ultrasound study found 31 cases in as survey of 100,000 the prevalence is 31/100,000 = 0.000031 or 0.0031%. If they are reporting incidence it should be the (average) annual incidence, which if it is 31.62 cases/100,000 in 27 years (as implied by the text) then the mean annual incidence is 1.17 cases per 100,000. Since the cases were based on hospital records of patients (as described in materials and methods) presenting for treatment, then the data is almost certainly incidence data and hence should be reported as an annual incidence of 0.49-1.17 cases / 100,000, Thus the annual incidence of 0.49-1.17 cases/100,000 in the high altitude mountainous areas was higher than the annual incidence of 0.180-0.282 cases per 100,000 seen in the low level areas. Likewise Mongolian (annual incidence 1.987/100,000) and Kazakh (annual incidence 0.897/100,000) had a higher incidence…...

It is extremely important to use standard epidemiological terms so that studies can be easily compared and trends in data seen. The standardized means of reporting numbers of human cases of any disease is annual incidence per 100,000 population. 

In the results section they have reported “annual prevalence” in some cases. This is not prevalence data, it is annual incidence, again correct the terminology. Likewise it is trend in incidence NOT prevalence. They have no prevalence data in humans in this study. They have correctly reported prevalence data in small mammals (ie proportion of small mammals infected).

Sincerely,

Paul Robert Torgerson

Associate Editor

Mar Siles-Lucas

Deputy Editor

The reviewers have found the revision largely satisfactory. However, before final acceptance of this manuscript, the authors need to ensure that reporting of population data with regard to diseases is standardized and using the correct epidemiological terms. For human cases is incidence per 100,000 per year. Please ensure all human incidence data is reported as such. From the materials and methods it seems the “cumulative prevalence” was based on the period 1989-2015, thus over a period of 27 years. This is confusing when making comparisons to other studies and gives a false impression that the incidence is somewhat higher than it is in reality.

In the author summary the way prevalence is reported is just wrong. Prevalence is the proportion with the disease at one point in time...ie if an ultrasound study found 31 cases in as survey of 100,000 the prevalence is 31/100,000 = 0.000031 or 0.0031%. If they are reporting incidence it should be the (average) annual incidence, which if it is 31.62 cases/100,000 in 27 years (as implied by the text) then the mean annual incidence is 1.17 cases per 100,000. Since the cases were based on hospital records of patients (as described in materials and methods) presenting for treatment, then the data is almost certainly incidence data and hence should be reported as an annual incidence of 0.49-1.17 cases / 100,000, Thus the annual incidence of 0.49-1.17 cases/100,000 in the high altitude mountainous areas was higher than the annual incidence of 0.180-0.282 cases per 100,000 seen in the low level areas. Likewise Mongolian (annual incidence 1.987/100,000) and Kazakh (annual incidence 0.897/100,000) had a higher incidence…...

It is extremely important to use standard epidemiological terms so that studies can be easily compared and trends in data seen. The standardized means of reporting numbers of human cases of any disease is annual incidence per 100,000 population. 

In the results section they have reported “annual prevalence” in some cases. This is not prevalence data, it is annual incidence, again correct the terminology. Likewise it is trend in incidence NOT prevalence. They have no prevalence data in humans in this study. They have correctly reported prevalence data in small mammals (ie proportion of small mammals infected).

Reviewer's Responses to Questions

**Key Review Criteria Required for Acceptance?**

**Methods**

-Are the objectives of the study clearly articulated with a clear testable hypothesis stated?

-Is the study design appropriate to address the stated objectives?

-Is the population clearly described and appropriate for the hypothesis being tested?

-Is the sample size sufficient to ensure adequate power to address the hypothesis being tested?

-Were correct statistical analysis used to support conclusions?

-Are there concerns about ethical or regulatory requirements being met?

Reviewer #1: (No Response)

Reviewer #2: (No Response)

**Results**

-Does the analysis presented match the analysis plan?

-Are the results clearly and completely presented?

-Are the figures (Tables, Images) of sufficient quality for clarity?

Reviewer #1: (No Response)

Reviewer #2: (No Response)

**Conclusions**

-Are the conclusions supported by the data presented?

-Are the limitations of analysis clearly described?

-Do the authors discuss how these data can be helpful to advance our understanding of the topic under study?

-Is public health relevance addressed?

Reviewer #1: (No Response)

Reviewer #2: (No Response)

**Editorial and Data Presentation Modifications?**

Reviewer #1: (No Response)

Reviewer #2: (No Response)

**Summary and General Comments**

Reviewer #1: (No Response)

Reviewer #2: Please clarify if human data were collected from 1989-2015 or 1989-2016. The end date varies throughout the abstract and text. 

Table 5- change infection rate to infection prevalence.

PLOS authors have the option to publish the peer review history of their article (what does this mean?). If published, this will include your full peer review and any attached files.

Reviewer #1: No

Reviewer #2: No
---

## [Editor Report · Decision Letter 3]

5 Oct 2020

Dear Dr. Zhang,

Thank you very much for submitting your manuscript "High endemicity of alveolar echinococcosis in Yili Prefecture, Xingjiang Autonomous Region, the People’s Republic of China: the infection situation of the disease in different ethnic communities and in small mammals" for consideration at PLOS Neglected Tropical Diseases. As with all papers reviewed by the journal, your manuscript was reviewed by members of the editorial board and by several independent reviewers. In light of the reviews (below this email), we would like to invite the resubmission of a significantly-revised version that takes into account the reviewers' comments. 

The authors were requested to report the incidence of reported cases in the standard format of cases per 100,000 per year. This they have failed to do, indeed there seems to be little effort to address this concern. As previously stated, this is very important top avoid confusion and misrepresentation of the epidemiology of this disease. Citing a cumulative incidence for example of 53.67/100,000 for example, will be interpreted as 53.67/100,000 per year (if the small print is not read or understood). There has been a modest attempt in the results section to include the data as annual incidence per 100,000, but this has not been included in the abstract and summary. Assuming this represents the number of cases over the26 years of the study period, then the mean annual incidence is 2.1 cases per 100,000 per year, which is typical of incidence in highly endemic regions. Also, since there were only 286 cases reported in this period from the relevant counties in Yili prefecture over 26 years. This is not a problem, but then converting it to cumulative incidence in the manner the authors have, give a false impression as to the extent of the disease. This manuscript should not be published until they have standardized the incidence data as cases per 100,000 per year throughout the manuscript and removed the emphasis on cumulative incidence.

We cannot make any decision about publication until we have seen the revised manuscript and your response to the reviewers' comments. Your revised manuscript is also likely to be sent to reviewers for further evaluation.

Sincerely,

Paul Robert Torgerson

Associate Editor

Mar Siles-Lucas

Deputy Editor

The authors were requested to report the incidence of reported cases in the standard format of cases per 100,000 per year. This they have failed to do, indeed there seems to be little effort to address this concern. As previously stated, this is very important top avoid confusion and misrepresentation of the epidemiology of this disease. Citing a cumulative incidence for example of 53.67/100,000 for example, will be interpreted as 53.67/100,000 per year (if the small print is not read or understood). There has been a modest attempt in the results section to include the data as annual incidence per 100,000, but this has not been included in the abstract and summary. Assuming this represents the number of cases over the26 years of the study period, then the mean annual incidence is 2.1 cases per 100,000 per year, which is typical of incidence in highly endemic regions. Also, since there were only 286 cases reported in this period from the relevant counties in Yili prefecture over 26 years. This is not a problem, but then converting it to cumulative incidence in the manner the authors have, give a false impression as to the extent of the disease. This manuscript should not be published until they have standardized the incidence data as cases per 100,000 per year throughout the manuscript and removed the emphasis on cumulative incidence.
---

## [Editor Report · Decision Letter 4]

15 Oct 2020

Dear Dr. Zhang,

We are pleased to inform you that your manuscript 'High endemicity of alveolar echinococcosis in Yili Prefecture, Xingjiang Autonomous Region, the People’s Republic of China: infection status in different ethnic communities and in small mammals' has been provisionally accepted for publication in PLOS Neglected Tropical Diseases.

Best regards,

Paul Robert Torgerson

Associate Editor

Mar Siles-Lucas

Deputy Editor

---

## [Editor Report · Acceptance letter]

13 Jan 2021

Dear Dr. Zhang,

We are delighted to inform you that your manuscript, " High endemicity of alveolar echinococcosis in Yili Prefecture, Xingjiang Autonomous Region, the People’s Republic of China: infection status in different ethnic communities and in small mammals," has been formally accepted for publication in PLOS Neglected Tropical Diseases.

Best regards,

Shaden Kamhawi

co-Editor-in-Chief

Paul Brindley

co-Editor-in-Chief
